# Cellular imbalance of specific RNA-binding proteins associates with harmful R-loops

**José Antonio Mérida-Cerro**[1¤], **Guillaume Chevreux**[2], **Benoit Palancade**[2], **Ana G. Rondón**[1,3]*, **Andrés Aguilera**[1,3]

**1** Centro Andaluz de Biología Molecular y Medicina Regenerativa (CABIMER), Universidad de Sevilla-CSIC-Universidad Pablo de Olavide, Seville, Spain, **2** Université Paris Cité, CNRS, Institut Jacques Monod, Paris, France, **3** Departamento de Genética, Facultad de Biología, Universidad de Sevilla, Seville, Spain

¤ Current Address: Centro de Investigación Médica Aplicada (CIMA), Universidad de Navarra, Pamplona, Spain

* agarcia13@us.es

## Abstract

Understanding how the assembly of nascent mRNA into a ribonucleoprotein (mRNP) influences R-loop homeostasis is crucial for gaining insight into the cellular mechanisms that prevent genome instability. Here, we identify three RNA-binding proteins, Rie1, Rim4 and She2, whose expression levels are important to limit R-loop accumulation and, thus, to prevent DNA damage. Interestingly, Rim4 and She2 are overrepresented in CBP80-containing mRNPs formed in the absence of THO. In addition, we found that an excess of the RNA exosome component Dis3 impairs its function, promoting R-loops, particularly from non-coding RNAs, which cause genomic instability. Our results indicate that changes in the availability of different RBPs or RNAs, causes R-loop-mediated DNA damage in the cell. These results may help to understand the mechanism that promotes cancer, as several RBPs are overexpressed in different types of tumors.

## Author summary

R-loops are three-stranded structures formed by a DNA-RNA hybrid and the displaced ssDNA. The accumulation of R-loops damages the genome, highlighting the need for their tight regulation. One of the mechanisms that prevents R-loop formation is the co-transcriptional assembly of the nascent RNA into a messenger ribonucleoprotein (mRNP). Our study reveals that not only the absence of mRNP components, but also the overexpression of specific RNA-binding proteins (RBPs) or the persistence of unstable transcripts, results in R-loop accumulation and consequent DNA damage. R-loops are involved in a number of neurodegenerative diseases and cancer, highlighting the importance of understanding the cellular processes that mitigate their accumulation.

**Data availability statement:** The authors confirm that all data underlying the findings are fully available without restriction. All relevant data are within the paper and its Supporting Information files. The mass spectrometry proteomics data have been deposited to the ProteomeXchange Consortium via the PRIDE partner repository with the dataset identifier PXD057248.

**Funding:** Research was funded by the following grants: PID2019-104270GB-I00 funded by Ministerio de Ciencia e Innovación/Agencia Estatal de Investigación/10.13039/501100011033 and FIUS22/0178 from Fundación de Investigación de la Universidad de Sevilla to AA; P18-FR-655 funded by Consejería de Universidad, Investigación de la Junta de Andalucía and by the European Union to AA and AGR; PP2024/00000588 funded by the University of Seville to AGR; ANR-21-CE12-0040 funded by Agence Nationale pour la Recherche to BP and PJA2022070005310 funded by Fondation ARC pour la recherche contre le Cancer to BP. The funders had no role in study design, data collection and analysis, decision to publish, or preparation of the manuscript.

**Competing interests:** The authors have declared that no competing interests exist.

## Introduction

R-loops are non-B DNA structures formed by a DNA-RNA hybrid and the displaced single-stranded DNA (ssDNA). Although R-loops appear naturally in cells and play important roles in physiological processes like class-switch recombination, mitochondrial DNA replication and telomere maintenance, their accumulation can lead to genome instability by interfering with essential processes such as transcription, replication, chromosome segregation and DNA damage repair [1–5]. In order to preserve their genomes, cells have developed several mechanisms to avoid R-loop formation and to resolve them. R-loops are prevented by the combined action of topoisomerases, which remove the negative supercoiling accumulated behind RNA polymerases (RNAPs), and the cotranscriptional assembly of nascent RNA into a mature messenger ribonucleoprotein (mRNP). Both activities reduce the capacity of RNA to tether back to DNA outside the RNAP, either by restraining DNA melting or by reducing RNA availability. Once formed, R-loops are resolved either by RNase H-mediated degradation of the RNA moiety or by the unwinding of the DNA-RNA hybrid through the action of multiple DNA-RNA helicases [5,6]. Defects in these processes might contribute to the development of cancer and human neurodegenerative diseases associated with the accumulation of R-loops [7].

The impact of mRNP assembly on R-loop homeostasis was initially described in mutants of the THO complex, a major mRNP component, and later extended to splicing factors and other mRNP factors [8]. THO was initially identified in yeast as a complex formed by Tho2, Hpr1, Mft1 and Thp2 [9] later on expanded to include Tex1 [10]. Structural analyses revealed that in yeast THO is a dimer [10–12]. During mRNP assembly, THO binds the DEAD-box helicase Sub2 and the mRNA export adaptor Yra1 [11–13], contributing to mRNP compaction and facilitating mRNA export [11,14]. Sub2 ATPase activity is enhanced through its interaction with THO [15]. Based on the phenotypes observed in *sub2* mutants and in cells depleted of its human counterpart, UAP56, removal of DNA-RNA hybrids formed co-transcriptionally is one of the many functions of Sub2 [16,17]. The role of THO in assembling the mRNP seems to be conserved in humans [14]. It has been hypothesized that mRNPs formed in the absence of THO would be less compacted, rendering the nascent RNA more susceptible to reannealing with DNA to form an R-loop [18,19].

The accumulation of DNA-RNA hybrids is not only caused by the absence but also by the excess of certain RNA-binding proteins (RBPs). Thus, elevating the cellular concentration of the RNA-binding factor Yra1 increases R-loops, probably due to the stabilization of the hybrids, given that Yra1 binds to DNA-RNA duplexes *in vitro* and it is recruited to genomic regions with R-loops *in vivo* in an RNase H-sensitive manner [20]. Additionally, Npl3, a factor involved in mRNP assembly and export that was previously reported to prevent R-loop formation [21], has recently been shown to bind and stabilize TERRA R-loops at short telomeres [22]. Therefore, RBPs could influence R-loops homeostasis either inhibiting the formation of DNA-RNA hybrids or stabilizing them once formed.

Despite the recent advances in elucidating mRNP composition and structure, the specific changes within mRNPs responsible for DNA-RNA hybrids formation remain unclear. To understand how mRNP assembly restricts R-loops, we analysed

Cbp80-containing mRNPs formed in the absence of the THO complex and found an increased presence of various RBPs and nuclear pore components. Moreover, we noticed that overexpressing the RBPs Rie1, Rim4 and She2 or altering the activity of the Dis3 RNA exosome subunit promotes R-loop accumulation in the cell, consequently causing DNA damage. Therefore, proper cellular expression levels of RBPs are crucial for preventing R-loop-mediated genome instability.

## Results

### mRNP proteome is altered in *hpr1* cells

To investigate the impact of mRNP biogenesis on R-loop homeostasis, we examined how mRNP composition changes in cells that accumulate R-loops. We chose *hpr1Δ* as a representative R-loop accumulating mutant and we undertook a quantitative proteomic approach to compare mRNP composition in the absence (*hpr1Δ*) or presence (WT) of the THO complex. We isolated mRNPs from WT or *hpr1Δ* cells by pulling down a tagged version of Cbp80, a component of the nuclear cap-binding complex (CBC), under conditions that preserve mRNP integrity [23]. Immunoprecipitates were analyzed by SDS-PAGE stained with a fluorescent dye and by mass spectrometry (Figs 1A and S1A). We have previously reported that the proteome of CBP pull-downs is consistent with the capture of nuclear mRNPs, as evidenced by the RNA-dependent association of most regular mRNP components, as assessed both by mass spectrometry analysis and western blotting upon RNase A treatment [24]. Consistently, our proteomic analysis detected a wide range of nuclear RNA-binding proteins: spliceosome components, RNA 3'-end processing and export factors, and RNA helicases among others (Fig 1A and S1 Table). Although the profile and intensity of the bands in the denaturing gel did not change significantly between WT and *hpr1Δ* (S1A Fig), the quantitative proteomic analysis identified consistent changes. First, mRNPs isolated from *hpr1Δ* cells are depleted not only of this subunit but of the entire THO complex, confirming that deletion of one subunit disassembled the whole complex [25] (Fig 1A). Interestingly, Sub2 is not reduced, suggesting that although THO might contribute to Sub2 loading, it might be able to bind independently or with the help of other factors that operate in the absence of THO. This idea is supported by Sub2 overexpression suppressing the defects in *tho* mutants [16]. Among the differences found in the mRNP composition of *hpr1Δ* cells, we noticed an increase in nuclear pore components (Nsp1, Seh1 and Nup57) and RBPs (Rim4, She2, Yra1, Sbp1 and Npl3). The increased binding to nucleoporins is consistent with the previously reported accumulation of mRNPs at nuclear pores in the absence of THO [26,27] and agrees with R-loops being redirected to the nuclear periphery [26].

Yra1 expression is tightly regulated in the cell, and its overexpression is toxic [28,29], especially in mutants that accumulate R-loops [20]. Indeed, *YRA1* overexpression causes R-loop-mediated DNA damage, which is detected as an increase in RNase H-sensitive Rad52 foci [20,30]. As Yra1 is overrepresented in *hpr1Δ* mRNPs (Fig 1A), we examined whether the other enriched RPBs, Rim4, She2, Sbp1, and Npl3, could also affect R-loop homeostasis. To do so, we first assessed *hpr1Δ* cell fitness when overexpressing each of these RBPs individually. We observed that only *SHE2* overexpression caused a mild growth defect in *hpr1Δ*. *RIM4* and *SBP1* overexpression are already toxic in wild-type cells complicating the detection of a stronger growth defect in *hpr1Δ* (Fig 1B). Next, we measured the accumulation of Rad52 foci as a readout of DNA damage. Interestingly, overexpression of *SHE2* and *RIM4* significantly increased Rad52 foci, similarly to *YRA1* (Fig 1C). We confirmed that although Npl3 and Sbp1 do not increase DNA damage, they are overexpressed to a similar level as She2 or Rim4, respectively (S1B and S1C Fig). When RNase H was simultaneously overexpressed, DNA damage was reduced to wild-type levels (Fig 1D) although the level of Rim4 or She2 remain unchanged (S1D Fig). Taken together, these results suggest that an excess of She2 and Rim4 in the mRNP or within the cell might disrupt R-loop homeostasis.

### Upregulation of four different RBPs induces genotoxic R-loops

Given the increase of certain RBPs in mRNPs assembled without the THO complex, along with the fact that several of them cause RNase H-sensitive DNA damage when overexpressed (Fig 1D), we asked whether overexpressing other

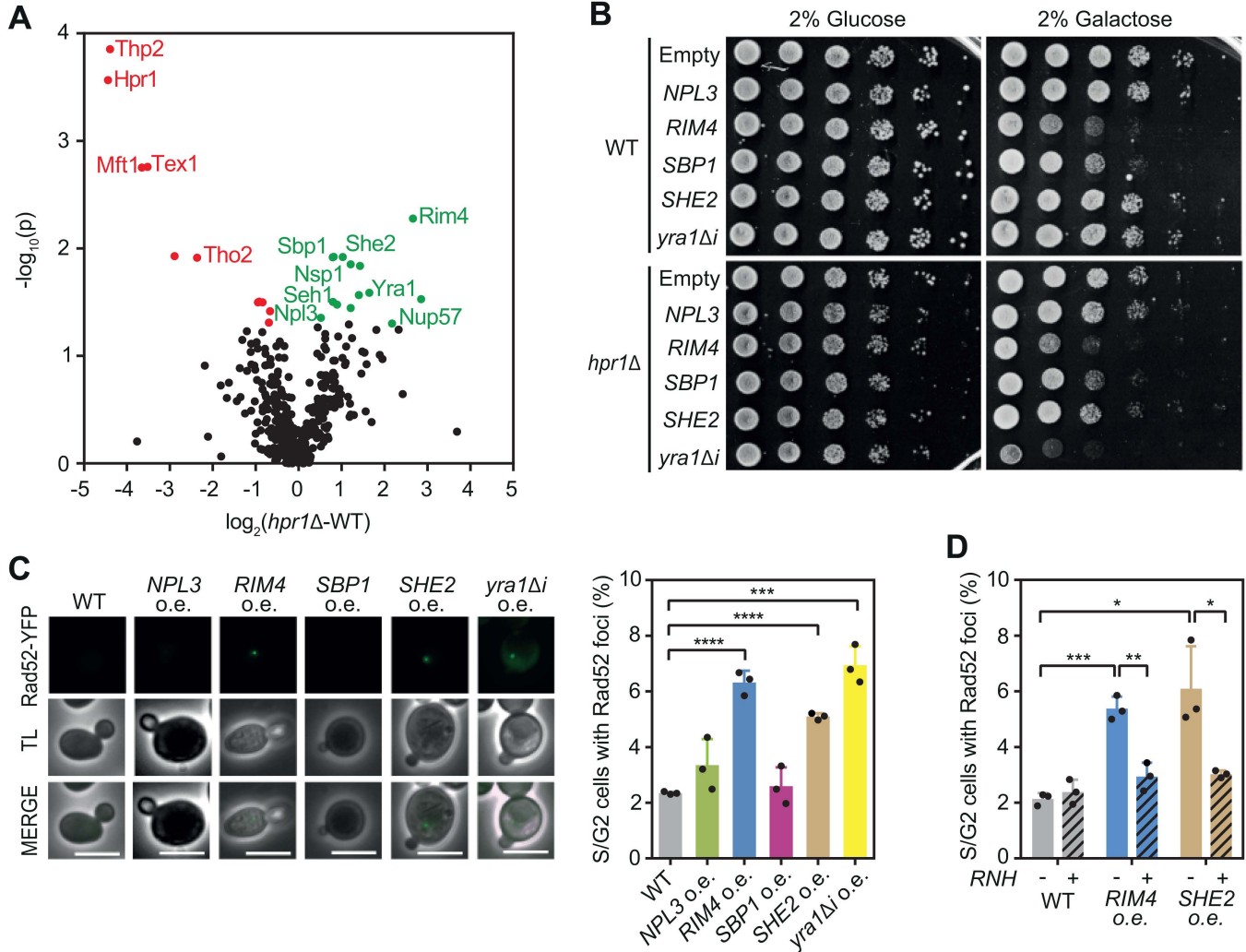

**Fig 1. Changes in RBPs leads to R-loop-dependent genome instability.** (A) Volcano plot of mRNP proteins in a *hpr1Δ* strain (H) (YAR015-4C) compared with a WT (W)(YDZ7005-16). Each dot represents a protein, with red indicating significant reduction and green indicating significant enrichment in *hpr1Δ* compared to WT. (B) Sensitivity of WT (YBP249) and *hpr1Δ* (HPBAR1-R) strains to the overexpression of *NPL3* (pYES-NPL3); *RIM4* (pYES-RIM4), *SBP1* (pYES-SBP1), *SHE2* (pYES-SHE2) or YRA1 (pYES-YRA1Δi). Cells were plated in SC media containing either 2% glucose or 2% galactose in order to repress or induce the overexpression. (C) Percentage of S/G2 cells with Rad52-YFP foci in a wild type (YBP249) overexpressing the indicated proteins. Representative images are shown. (D) Percentage of S/G2 cells with Rad52-YFP foci in a wild type (YBP249) overexpressing RIM4 (*RIM4* o.e.), or SHE2 (*SHE2* o.e.) combined (+) or not (-) with the overexpression of RNase H. Fold change and p-value mean of 4 samples are plotted for (A). Mean and SD of 3 samples are plotted for (C) and (D). *p ≤ 0.05; **p ≤ 0.01; ***p ≤ 0.001; ****p ≤ 0.0001 (two-tailed Student's t-test).

RBPs could cause R-loop accumulation. To address this, we performed a screen for genes that reduce growth when overexpressed in *hpr1Δ* cells, a phenotype previously described for one of these RBPs, Yra1 [20] and observed for She2 (Fig 1B). We used a diploid *hpr1Δ/HPR1-AID* strain expressing a degron version of *HPR1* that induces Hpr1 depletion through auxin addition. We transformed it with the MW90 multicopy library and examined its growth in the absence of Hpr1. Overexpression of *YRA1* (Yra1Δi) was included as positive control [20]. We isolated two plasmids that caused a mild growth defect: clones 1 (C1) and 2 (C2) (Fig 2A). Next, we assessed whether expression of clone 1 or 2 increased R-loop-dependent DNA damage by quantifying Rad52 foci in cells overexpressing or not RNase H. Both clones increased Rad52

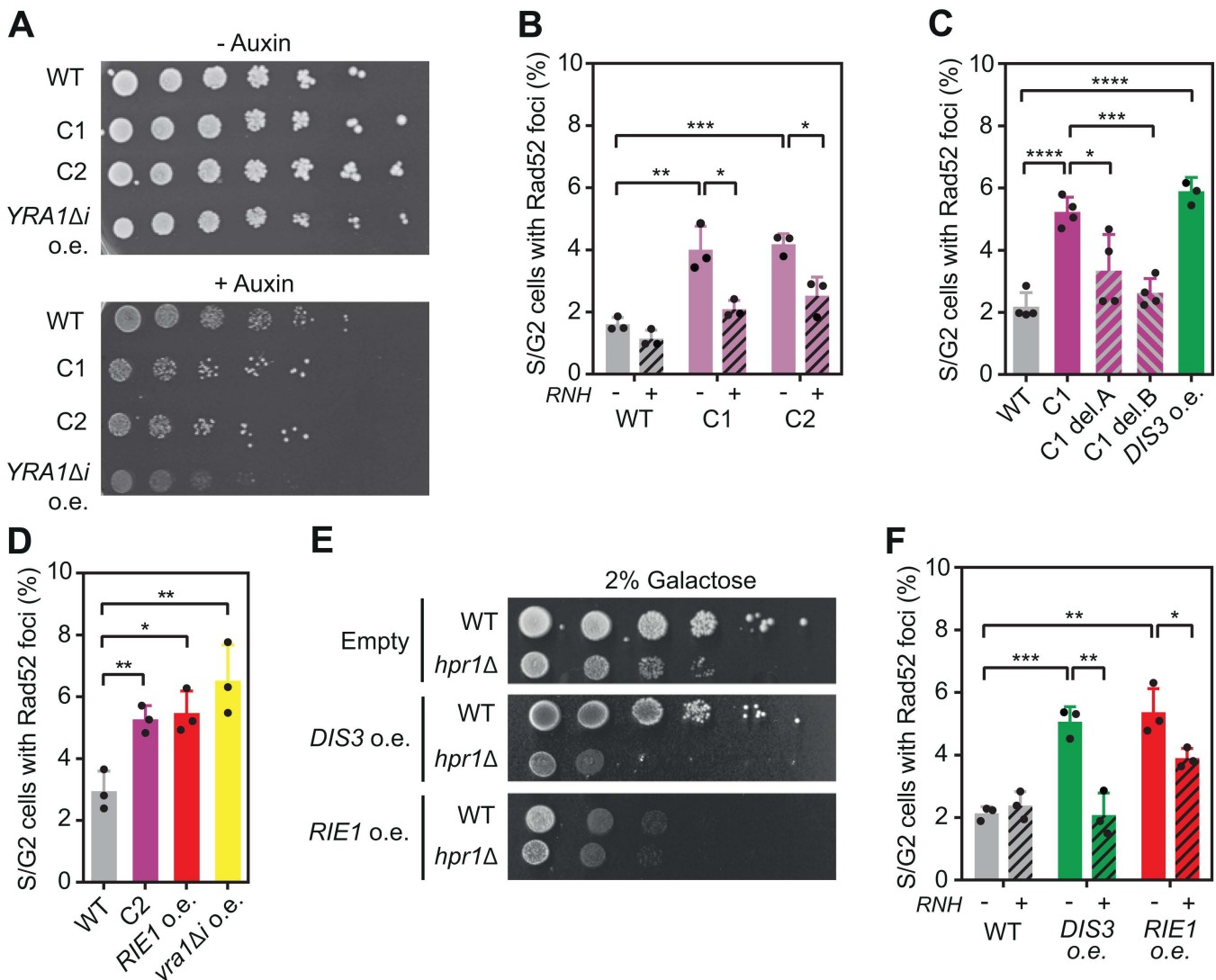

**Fig 2. *DIS3* and *RIE1* upregulation induces R-loop-dependent DNA damage.** (A) Growth of diploid hpr1/HPR1AID (HPR1DGK) strain Hpr1 depleted (+Auxin) or non-depleted (-Auxin) transformed with YEp351 (WT), MW90-clone1 (C1) or MW90-clone2 (C2), YEpYRA1Δi (*yra1Δi*). (B) Percentage of S/G2 cells with Rad52-YFP foci in a wild-type (YBP249) strain overexpressing C1 or C2, combined (+) or not (-) with overexpression of RNase H. (C) Percentage of S/G2 cells with Rad52-YFP foci in a wild type strain (YBP249) overexpressing the indicated clones or DIS3 (*DIS3* o.e.). (D) Percentage of S/G2 cells with Rad52-YFP foci in a wild type strain (YBP249) overexpressing C2, RIE1 (*RIE1* o.e.) or YRA1 (*yra1Δi*). (E) Sensitivity of WT (YBP249) and *hpr1Δ* (HPBAR1-R) to elevated levels of DIS3 (*DIS3* o.e.) or RIE1 (*RIE1* o.e.). (F) Percentage of S/G2 cells with Rad52-YFP foci in a wild type strain (YBP249) overexpressing *DIS3* or *RIE1*, combined (+) or not (-) with overexpression of RNase H. Mean and SD of ≥3 samples are plotted for (B), (C), (D) and (F). *p ≤ 0.05; **p ≤ 0.01; ***p ≤ 0.001; ****p ≤ 0.0001 (two-tailed Student's t-test).

foci similarly to *YRA1* overexpression, and this effect was suppressed by RNase H overexpression, consistent with an accumulation in DNA-RNA hybrids (Fig 2B). The first clone (C1) contained a fragment of chr XV including two tRNA genes (*SUF17* and *SUP3*), *TAT2*, *DIS3* and the putative gene *YOL019W-A* (S2A Fig). Interestingly, Dis3 is an endo-exonuclease that forms part of the RNA exosome, a complex that degrades RNA [31]. To assess whether overexpression of *DIS3*, but not the other genes cloned in C1, is detrimental to *hpr1* cells, we truncated the C1 clone by removing *SUF17* and part of (C1del.A) or the entire *DIS3* gene (C1del.B) (S2A Fig). Neither C1del.A nor C1del.B expression caused DNA damage.

However, overexpressing *DIS3* from a multicopy plasmid increased Rad52 foci (S2C Fig and Fig 2C) and caused cell sickness in *hpr1Δ* cells (Fig 2E). We concluded that an excess of Dis3, rather than the tRNAs or Tat2 protein, is responsible for *hpr1Δ* sickness.

The other clone identified (C2) had a 7.2 kb insert of chr VII including *MGA1* and *RIE1* (YGR250c) genes (S2B Fig). Interestingly, Rie1 is an RNA-binding factor with three RNA recognition motifs (RRMs). Thus, we examined whether *RIE1* overexpression was responsible for the increase in DNA damage. An excess of Rie1 increased Rad52 foci in the cells to a similar level than C2 plasmid (S2C and 2D Figs) and strongly decreased fitness in wild-type cells, making difficult to observe any additional effect in *hpr1Δ* cells (Fig 2E). When we overexpressed RNase H, *DIS3* or *RIE1* expression was not affected but Rad52 foci decreased (S2D and 2F Figs), suggesting that an excess of Dis3 or Rie1 alters DNA-RNA hybrid homeostasis in the cell. Consistent with these results, Rie1 is overrepresented in Hpr1-depleted mRNPs and, although Dis3 was not detected in the proteomic analysis, several RNA exosome subunits preferentially bind to mRNPs in the absence of THO (S1 Table). Taken together the results obtained in the proteomic and the genetic approaches, we identified four RBPs that cause R-loop dependent DNA damage when overexpressed: Dis3, Rie1, Rim4 and She2. Therefore, not only the removal of mRNP components but also their overexpression increases genomic damage, most probably favoring R-loop accumulation.

To directly examine whether overexpression of *DIS3*, *RIE1*, *RIM4* or *SHE2* increases R-loops, we performed an immunofluorescence assay in yeast chromosome spreads using the S9.6 antibody, which recognizes DNA-RNA hybrids and with less affinity dsRNA [32]. We observed DNA-RNA hybrids in 17–21% of the cells upon overexpression of the genes analyzed, compared to 9% without it. Simultaneous overexpression of RNase H *in vivo* reduced the S9.6 signal, confirming the specificity of the signal for DNA-RNA hybrids (Fig 3A). We then investigated the localization of these R-loops in the genome by DNA-RNA immunoprecipitation (DRIP) followed by qPCR at *GCN4* and ribosomal DNA 18S, two regions previously validated for R-loop detection in yeast [20]. We only observed a significant enrichment of hybrids in these regions upon Rie1 or Rim4 upregulation (Fig 3B), but as Dis3 and She2 overexpression also increased the S9.6 signal in the immunofluorescence experiments (Fig 3A), which provides a global view of R-loop accumulation in the genome, we tested other genomic regions functionally connected to these factors. For Dis3, we looked at *NLR021w*, *NEL025c*, and *NHR027c* genes encoding cryptic unstable transcripts (CUTs) that are degraded by the RNA exosome [33], whereas for She2 we examined *ASH1, EAR1* and *IST2* genes whose transcripts contain RNA secondary structures specifically recognized by She2 [34,35]. We observed a significant accumulation of DNA-RNA hybrids in CUTs encoding genes when Dis3 is in excess (Fig 3C) and in genes whose transcripts are recognized by She2 when this RBP is overexpressed (Fig 3D), confirming that R-loops also accumulate upon overexpression of these factors. Taken together, our results imply that altering the cellular levels of Rim4, Rie1, She2 or the RNA exosome subunit Dis3, causes R-loop accumulation. Interestingly, whereas Rim4 and Rie1 seem to have a general effect, She2 and Dis3 action is preferentially directed to specific genes whose transcripts are targets of these factors.

## Cellular dysfunction of Dis3 and Rim4 induce harmful R-loops

The expression of each protein in the cell is optimized to favor specific interactions and to preserve the stoichiometry of the complexes. Thus, we reasoned that the drastic increase caused by expressing these RBPs from multiple copies could be disrupting their activity. To test this idea, we assessed whether mutating *RIM4*, *SHE2*, *DIS3* or *RIE1* would increase DNA-RNA hybrids, as previously observed with their overexpression (Fig 3A). By immunofluorescence on chromosome spreads, we observed a significant increase in the percentage of nuclei with S9.6 signal in *dis3–1* (24%) and *rim4Δ* (27%) mutants, similar to the *rnh1Δ* mutant lacking RNase H1 (25%) (Fig 4A), and *DIS3* and *RIM4* overexpression (Fig 3A). In contrast, neither *she2Δ* nor *rie1Δ* deletion caused hybrid accumulation (Fig 4A). Consistent with the immunostaining data, we detected DNA damage, measured as Rad52 foci accumulation, in *dis3–1* and *rim4Δ* but not in *she2Δ* or *rie1Δ* cells (Fig 4B). These experiments suggest that expressing from a multicopy vector *DIS3* and *RIM4* might disrupt the function of these factors mimicking the *dis3* and *rim4Δ* mutations.

PLOS Genetics

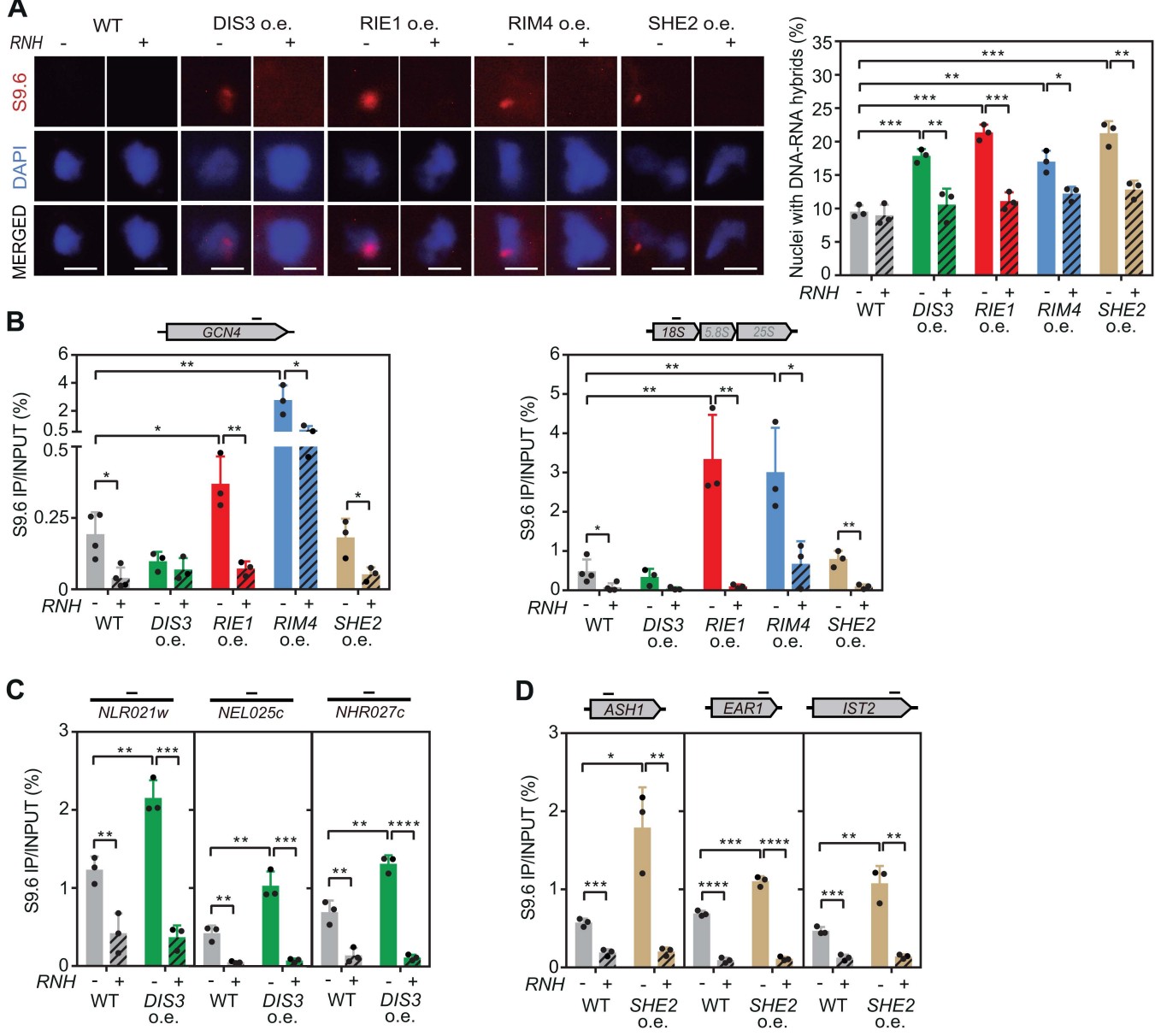

**Fig 3. Elevated *DIS3*, *RIE1*, *RIM4* or *SHE2* expression increases R-loops.** (A) Representative pictures and percentage of nuclei with S9.6 signal from a wild type strain (YBP249) overexpressing *DIS3*, *RIE1*, *RIM4* or *SHE2*, combined (+) or not (-) with overexpression of RNase H. Data from more than 100 total cells per experiment are shown. (B) DRIP using S9.6 antibody in *GCN4* and rDNA *18S* genes in YBP249 WT strain overexpressing *DIS3*, *RIE1*, *RIM4* or *SHE2* either non-treated (RNH-) or *in vitro* treated (RNH+) with RNase H. (C) DRIP in *NLR021w*, *NEL025c* and *NHR027c* CUTs in YBP249 overexpressing *DIS3* (*DIS3* o.e.) either non-treated (RNH-) or *in vitro* treated (RNH+) with RNase H. (D) DRIP in *ASH1*, *EAR1* and *IST2* genes in YBP249 overexpressing *SHE2* (*SHE2* o.e.) either non-treated (RNH-) or *in vitro* treated (RNH+) with RNase H. Mean and SD of 3 samples are plotted for (A), (B), (C) and (D). *p ≤ 0.05; **p ≤ 0.01; ***p ≤ 0.001; ****p ≤ 0.0001 (two-tailed Student's t-test).

Next, we addressed the mechanism by which Dis3 and Rim4 prevent R-loop accumulation. First, we confirmed that the increase in S9.6 signal observed in *dis3–1* and *rim4Δ* is indeed an increase in DNA-RNA hybrids by analyzing RNase H sensitivity. We observed that the number of nuclei with S9.6 signal was significantly reduced by *in vivo* RNase H overexpression (Fig 4C). Moreover, Rad52 foci were also partially suppressed by RNase H (Fig 4D).

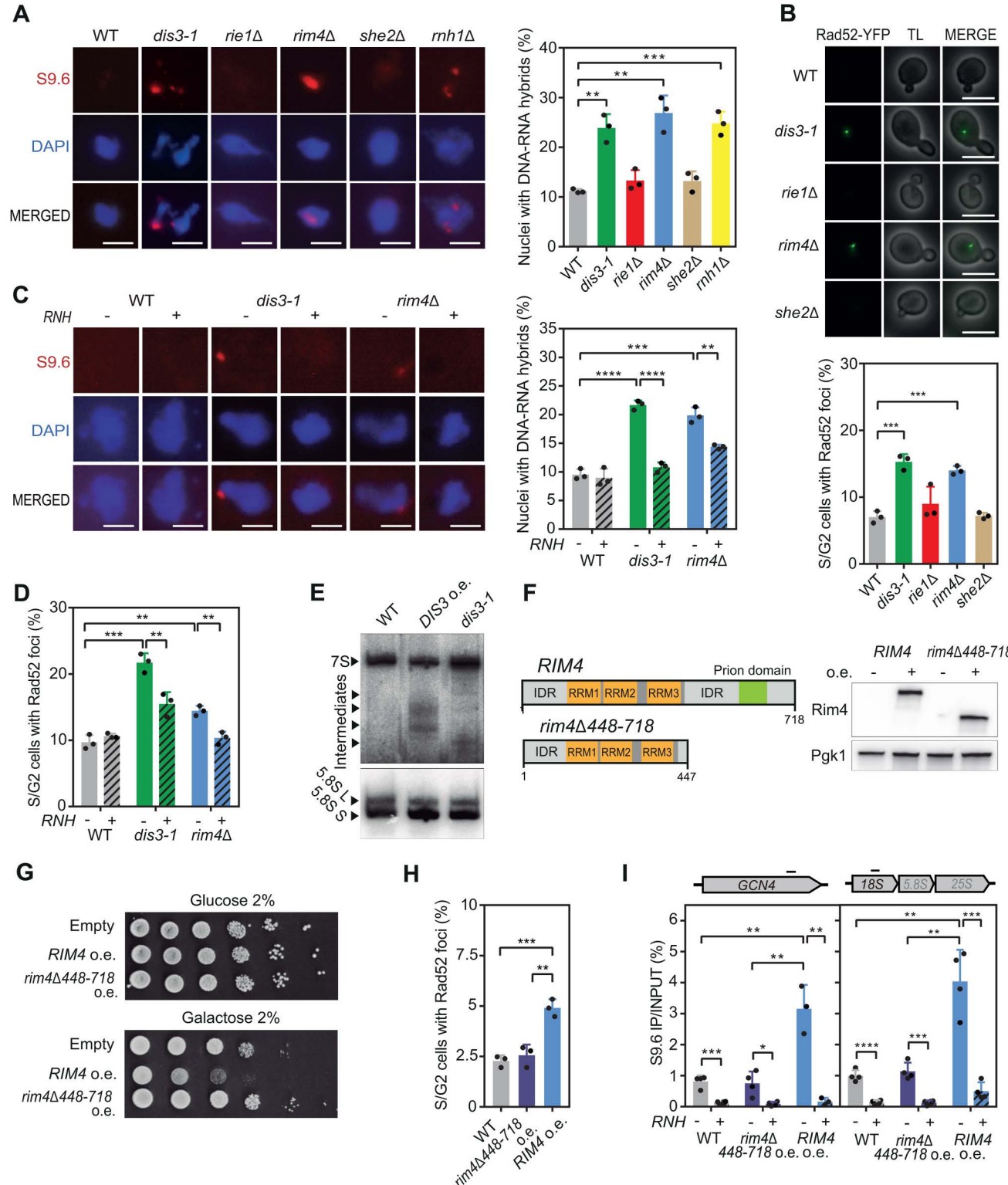

**Fig 4. Mutations and excess of Dis3 or Rim4 increases DNA damage.** (A) Representative pictures and percentage of nuclei with S9.6 signal from BY4741 (WT), BDIS3-1 (*dis3-1*), BRIE1D (*rie1Δ*), BRIM4D (*rim4Δ*), BSHE2D (*she2Δ*) and BRNH1D (*rnh1Δ*) strains. (B) Percentage of S/G2 cells with Rad52-YFP foci in BY4741 (WT), BDIS3-1 (*dis3-1*), BRIE1D (*rie1Δ*), BRIM4D (*rim4Δ*) and BSHE2D (*she2Δ*) strains. Representative images are shown.

(C) Representative pictures and percentage of nuclei with S9.6 signal from BY4741 (WT), BDIS3-1 (*dis3-1*) and BRIM4D (*rim4Δ*) strains overexpressing (RNH+) or not (RNH-) RNase H. (D) Percentage of S/G2 cells with Rad52-YFP foci in BY4741 (WT), BDIS3-1 (*dis3-1*) and BRIM4D (*rim4Δ*) overexpressing (RNH+) or not (RNH-) RNase H. (E) Northern blot of 5.8S rRNA from BY4741 transformed with YEp351 (WT) or YEpDIS3 (*DIS3* o.e.) or BDIS-1 (*dis3-1*) strain. Unprocessed 7S rRNA, processed 5.8S L and S forms and intermediaries rRNAs are marked. (F) Diagram of *RIM4* and *rim4Δ488-718*. Lysates of a WT strain (YBP249) overexpressing (+) or not (-) Rim4 and Rim4Δ488-718 were analyzed by Western blot. (G) Sensitivity of a WT strain (YBP249) to overexpression of *RIM4* (pYES-RIM4) or the truncated *rim4Δ488-718* version (pYES-RIM4Δ488-718). (H) Percentage of S/G2 cells with Rad52-YFP foci in a wild type (YBP249) overexpressing *Rim4 or Rim4Δ488-718*. (I) DRIP using S9.6 antibody in *GCN4* (left) and rDNA *18S* (right) genes in a wild type (YBP249) overexpressing full-length (*RIM4*) or truncated *RIM4* (*rim4Δ488-718*) either non-treated (RNH-) or *in vitro* treated (RNH+) with RNase H. Mean and SD of ≥3 samples are plotted for (A), (B), (C), (D), (H) and (I). $*p \le 0.05$; $**p \le 0.01$; $***p \le 0.001$; $****p \le 0.0001$ (two-tailed Student's t-test).

Therefore, we confirmed that *dis3–1* and *rim4Δ* mutations caused R-loops accumulation in the cell leading to DNA damage.

Given that *DIS3* forms part of the RNA exosome [31], we wondered whether increasing *DIS3* expression in the cell was affecting the function of this complex. RNA exosome depletion was previously described to affect 7S rRNA 3'-end processing into mature 5.8S rRNA [36,37]. Therefore, we analyzed 5.8 rRNA maturation by Northern blot in cells overexpressing *DIS3* and compared it with the *dis3–1* mutant. In wild-type cells, we only detected the 7S precursor and the completely processed 5.8L and 5.8S forms. However, in the RNA exosome mutant or upon *DIS3* overexpression, we also observed intermediate forms of varying sizes (Fig 4E), confirming RNA exosome dysfunction and the stabilization of transient RNAs. We reasoned that expressing *DIS3* from a multicopy vector might overload the cell with this component, potentially disrupting RNA exosome function and stabilizing short-lived RNAs favoring DNA-RNA hybrids.

Rim4 contains three RNA recognition motifs at the N-terminal region and a prion domain at the C-terminal part of the protein, which is involved in formation of amyloid-like aggregates (Fig 4F) [38]. Because an excess of this protein causes similar phenotypes to the loss, we wondered whether the overexpression might trigger protein aggregation, thereby depleting the cell of functional Rim4. To test this idea, we overexpressed a truncated version of *RIM4*, missing the 271 C-terminal amino acids (*rim4Δ448–718*) (Fig 4F), previously reported not to form aggregates *in vivo* [38]. We confirmed that the truncated protein was overexpressed to the same level as the full-length protein (Fig 4F). However, removing the C-terminal IDR and prion domains restored cell viability (Fig 4G) and suppressed the DNA damage observed when the full-length protein is overexpressed (Fig 4H). DRIP analysis confirmed that whereas overexpression of the full protein increased DNA-RNA hybrids at *GCN4* and *18S rDNA*, overexpression of *rim4Δ448–718* presented similar levels to the wild-type strain in both genes (Fig 4I). The data suggest that an excess of Rim4 triggers its aggregation, causing a depletion of functional Rim4 in the cells.

Our data suggest that the increase of R-loops caused by an excess of Rim4 and Dis3 may result from the disruption of their physiological functions in the cell.

## Nuclear accumulation of She2 or Rie1 alters R-loop homeostasis

Finally, we sought to understand how overexpression of Rie1 and She2 could interfere with R-loop homeostasis, leading to their accumulation. Initially, we wondered whether She2 or Rie1 accumulate in the nucleus when present in excess. This is especially relevant for Rie1, previously described as a cytoplasmic protein [39], but also important for She2, which shuttles between both compartments [40]. We used strains with the endogenous copy of *SHE2* or *RIE1* tagged with YFP to analyze the localization of these RBPs under normal conditions and we transformed these strains with multicopy plasmids with YFP-tagged versions of *SHE2* or *RIE1* to overexpress them. As expected, we observed a significant increase in either Rie1 or She2 in cells transformed with the overexpression plasmids (Fig 5A). We quantified the YFP signal in cells overexpressing or not *SHE2* or *RIE1* and plotted the nucleus/cytoplasm

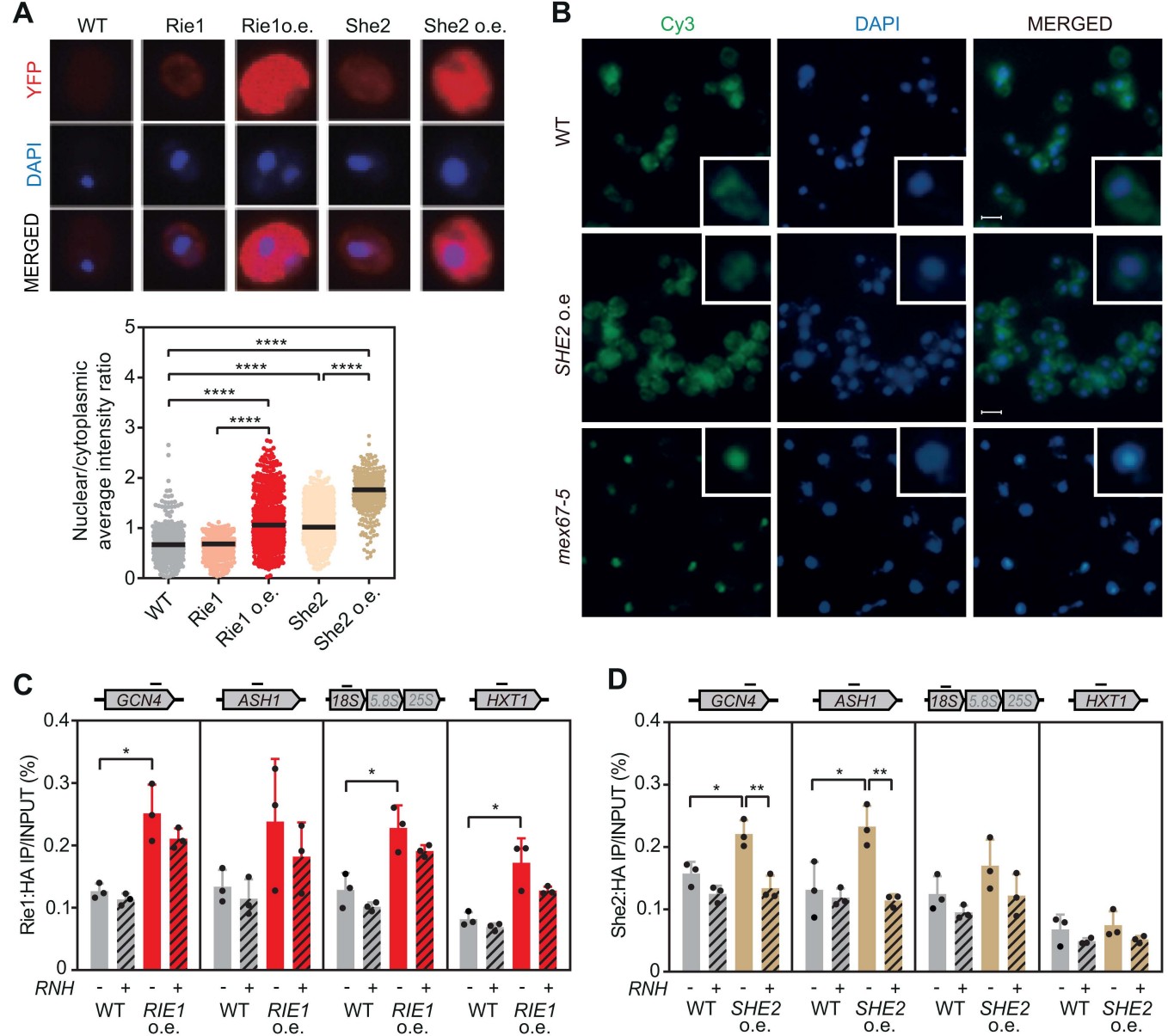

**Fig 5. Elevated expression of Rie1 or She2 increases their chromatin recruitment.** (A) *In vivo* localization of Rie1-YFP or She2-YFP in BY4741 overexpressing or not these factors. DNA was stained with DAPI. Ratio of the YFP average intensity detected in the nucleus and the cytoplasm. (B) Localization of poly(A)+ RNA by FISH with Cy3-labeled oligo(dT) in BY4741 overexpressing *SHE2* (*SHE2* o.e.) and in WMC1-1A strains (*mex67-5*). DNA was stained with DAPI. A detail of a single cell is highlighted. (C) ChIP using anti-HA antibody in WRIE1HA transformed with pYES2 (WT) or pYES-RIE1HA (*RIE1* o.e.) overexpressing (RNH+) or not (RNH-) RNase H. *GCN4*, *ASH1*, *18S* and *HXT1* genes were analysed. (D) As in (C) but in WSHE2HA strain with *SHE2-HA* overexpression. Individual values and median of 3 samples are plotted for (A) (two-tailed Mann-Whitney t-test). Mean and SD of 3 samples are plotted for (C) and (D) (two-tailed Student's t-test) *p ≤ 0.05; **p ≤ 0.01; ****p ≤ 0.0001.

ratio (Fig 5A). We confirmed that Rie1 localized predominantly in the cytoplasm (ratio< 1), but upon overexpression, we observed that it entered the nucleus becoming equally distributed in both compartments (ratio ≅ 1). Conversely, She2 was initially distributed throughout the cell (ratio ≅ 1), and tended to accumulate in the nucleus when present in excess (ratio >1) (Fig 5A).

Since we have previously reported that mutation in the major mRNA export factor Mex67 increases genomic instability [16], and considering that She2 is involved in transferring specific mRNAs to the yeast bud tip [41,42], we next investigated whether She2 accumulation in the nucleus could induce promiscuous interactions that affected general mRNA export. To explore this, we performed fluorescence *in situ* hybridization (FISH) using an oligo(dT)-Cy3 probe to detect polyadenylated transcripts. As a control, we included a mutant of *MEX67*. We observed that cellular mRNA distribution did not change upon *SHE2* overexpression, in contrast to control *mex67–5* cells (Fig 5B). Thus, despite accumulating in the nucleus, an excess of She2 does not alter bulk mRNA export to the cytoplasm.

A different possibility could be that the accumulation of Rie1 and She2 in the nucleus might cause aberrant binding to R-loops, as we previously observed with the overexpression of Yra1 [20]. To address this possibility, we overexpressed an HA-tagged version of Rie1 or She2 in strains where the endogenous copy is also tagged (S1C and S2C Fig). By Chromatin Immunoprecipitation analysis, we observed that Rie1 was recruited to chromatin when in excess. However, this interaction was not transcription-dependent as Rie1 was also bound to the repressed gene *HXT1*. In addition, the interaction of Rie1 with chromatin is R-loop-independent, as it persisted in the presence of RNase H (Fig 5C). In contrast, She2 localized to *GCN4* and *ASH1*, but not to *HXT1*, in a hybrid-dependent manner (Fig 5D), indicating that She2 accumulation in the nucleus may be at R-loop-forming loci, potentially stabilizing transient DNA-RNA hybrids.

Our observations suggest that R-loop homeostasis can be challenged by the increased nuclear localization of She2 and Rie1, which may induce aberrant interactions with either transiently formed R-loops or chromatin, ultimately favoring R-loop accumulation.

## Discussion

A major source of mRNP-dependent genome instability are unscheduled R-loops, which disrupt transcription and replication, causing DNA damage [2,6]. Our data provide evidence that an excess of the RNA exosome nuclease Dis3 or the RBPs Rim4, She2 and Rie1 can increase R-loops, leading to DNA damage. The mechanisms by which these proteins promote R-loop accumulation vary, highlighting the different cellular functions that control R-loop homeostasis. Thus, overproduction of Dis3 causes RNA exosome malfunction, resulting in the stabilization of short-life transcripts that may be more prone to hybridize with DNA. Alternatively, Rie1 or She2 overexpression increases their nuclear concentration, causing aberrant interactions with chromatin or nascent mRNPs that in the case of She2 are R-loop mediated. Finally, the amyloid-like protein Rim4, at physiological levels, seems to protect cells from R-loops through a yet uncharacterized mechanism, being more frequently recruited to R-loop prone mRNPs (Fig 1A). Our study, therefore, suggests that precise control of RBP availability is crucial for maintaining genome integrity. We propose that mRNP assembly could be challenged not only by specific RBP depletion, but also by its overexpression. Regardless of the mechanism, if mRNPs are not properly formed, R-loops tend to accumulate in the cell, leading to DNA damage.

The THO complex was the first mRNP component implicated in R-loop prevention, a role later extended to other RBPs [6]. As RNA-binding factors, we proposed that defective mRNP assembly might allow RNA to reanneal with template DNA [18,25]. To determine the nature of these putative aberrant mRNPs prone to forming R-loops, we analyzed their composition by quantitative proteomics, comparing mRNPs in the presence and absence of THO (Figs 1A and S1A). In addition to a marked reduction in the five THO subunits, which confirms they form a structural unit independent of Sub2 and Yra1, in agreement with recent findings [11], we observed a significant increase in several RBPs (Yra1, Npl3, Rim4, She2 and Sbp1) and in nucleoporins (Seh1, Nup57 and Nsp1) (Fig 1A). Several mechanisms could underlie the changes in the mRNP components observed in THO-complex mutants. The increase in RBPs observed could result from nonspecific binding to a less compacted mRNP. Alternatively, in *hpr1Δ* cells, mRNA release from the transcription site, co-transcriptional assembly with proteins, and export out of the nucleus may be kinetically affected so that different stages of nuclear mRNA metabolism are captured as compared to wild-type cells, in line with earlier studies [13,27]. Finally, higher levels of DNA-RNA hybrids in *hpr1Δ* cells may further impact the outcome of mRNP pull-outs, since chromatin-bound

nascent mRNPs may associate with hybrid-binding proteins. This later possibility could apply to Yra1 and She2, both of which are overrepresented in mRNPs without THO (Fig 1A). We have previously shown that Yra1 interacts with DNA-RNA hybrids *in vitro* and when present in excess in cells, binds to R-loop-prone genomic regions in an RNase H sensitive manner, increasing genome instability [20,30]. Increased Yra1 association with RNAs engaged in R-loops may account for its capture in *hpr1Δ* pull-downs (Fig 1A), possibly explaining the difference with a published report detecting decreased Yra1 levels in single molecule analyses of free mRNPs isolated from THO-complex mutant cells [43]. Similarly, overexpressed She2 is recruited to chromatin in a RNase H-sensitive manner and causes R-loop accumulation and DNA damage (Figs 1D, 3A, 3D and 5D) opening the possibility that She2 might bind to and/or stabilize unwanted DNA-RNA duplexes. In the case of the observed increase in nucleoporins, it may be a consequence of the previously reported interaction of THO-deficient mRNPs with the nuclear pore [27] or the recently described relocation of R-loops to the nuclear pores [26]. Therefore, although other RBPs may partially compensate for the absence of the THO complex in mRNPs, they may inadvertently promote the accumulation of DNA-RNA hybrids and their retention at the nuclear envelope.

The identification of Dis3 as a defense against R-loops has revealed another aspect of mRNP metabolism that may contribute to R-loop homeostasis. Interestingly, overexpression of *DIS3* alters the function of the RNA exosome, probably by unbalancing the stoichiometry of the complex, leading to accumulation of intermediates in rDNA maturation (Fig 4E). RNA exosome dysfunction, whether due to *DIS3* overexpression or mutation, stabilizes RNAs especially non-coding species, increasing the likelihood of RNA hybridizing to DNA. Consistent with this, the TRAMP component Trf4, which helps the RNA exosome to degrade RNAs, protects cells from R-loop-triggered genome instability and the lack of the nuclear RNA exosome nuclease Rrp6 or other TRAMP components increases genome instability [44–46]. The role of the RNA exosome in preventing R-loops is conserved in humans, where *DIS3* loss enhances R-loop mediated genome instability [47]. Moreover, increasing ncRNA synthesis, by human *SPT6* depletion, or its stabilization, by RNA exosome depletion, enhances R-loops [48,49]. This supports the idea that ncRNAs are prone to hybridizing with DNA if they are not rapidly degraded. However, other indirect mechanisms leading to R-loops cannot be excluded. For instance, RNA accumulation due to RNA exosome malfunction might deplete cells of RBPs, as recently reported for the Nrd1-Nab3-Sen1 complex involved in non-coding transcription termination [50]. Additionally, the absence of the RNA exosome could affect Senataxin, an DNA-RNA helicase involved in hybrid resolution, with which the RNA exosome interacts [51].

Interestingly, two factors that lead to the accumulation of R-loops when overexpressed, Rie1 and Rim4, contain intrinsically disordered regions (IDRs). These unstructured domains enable beneficial interactions with proteins and nucleic acids and were recently proposed to contribute to mRNP compaction [11,52]. However, when present at abnormal levels, they can promote undesirable contacts with non-physiological partners. This might be the case for Rie1, which, when in excess, enters the nucleus and binds to chromatin in a transcription and hybrid independent manner (Fig 5C), altering the chromatin environment in a yet uncharacterized way that enhances R-loops. In the case of Rim4, an important distinction is that, unlike Rie1, both mutation and overexpression lead to R-loop accumulation. Given that the IDR of Rim4 functions as a prion-like domain, capable of folding into amyloid-like aggregates [38], and that this prion-like domain is responsible for Rim4 toxicity and the accumulation of R-loops when Rim4 is overexpressed (Fig 4F-4I), we hypothesize that Rim4 accumulation might trigger its aggregation, which disrupt its normal function in preventing R-loops. There are precedents for this in humans, where TDP-43 and FUS, two RBPs with prion-like domains [53], also promote R-loop accumulation when mutated [54–56]. Overexpression of TDP-43 and mutations in FUS that lead to the formation of cytoplasmic aggregates, depleting the nucleus of functional protein, contributes to the neurodegenerative diseases amyotrophic lateral sclerosis (ALS) and frontotemporal dementia (FTD) [57,58]. Although it remains to be determined whether R-loops play a direct role in the development of these diseases, it would be interesting to know how IDR-containing RBPs influence R-loop homeostasis.

In this study, we found that an excess of She2 and Rie1, or lack of a functional RNA exosome or Rim4, increases DNA damage by enhancing R-loops. The mechanism underlying R-loop increment could vary, ranging from increased RNA

stability to promiscuous interactions that promote or stabilize R-loops. In line with these diverse mechanisms, several distinct factors contribute to prevent this stress, a list that is still increasing since THO was initially identified, and includes not only structural mRNP components like Npl3, but also processing factors like Rna14 or RNA helicases like Sen1 [6,45,59], and has now been expanded to the RNA exosome, She2 and the IDR-containing proteins Rie1 and Rim4. Taken together, these observations suggest that mRNP biogenesis is a delicate and precise process, and stressing it by removing or increasing specific factors, causes R-loop mediated DNA damage. This might be particularly relevant in natural situations like aging or diseases such as cancer where the level of multiple RBPs is altered, leading to cellular stress [60,61]. RBPs may thus represent a promising class of potential therapeutic targets, not only in line with their known roles in genome expression and alternative splicing, but also in view of their impact on R-loops and genetic stability. Modulating R-loop levels could be important in diseases associated with excess of these structures such as cancer, autoimmune disorders, and neurodegenerative diseases, as increasing R-loops by stressing mRNP biogenesis could result in cell death in combination with inhibition of the DNA damage response, in line with anti-cancer therapeutic approaches [62]. Our work, thus, supports the existence of an additional level of R-loop control that relies on an optimal balance of RBPs that could open new perspectives for therapeutic strategies for R-loop-associated diseases.

## Materials and methods

### Yeast strains and media

Yeast strains used in this study are derivatives of W303 (*MATa his3–11,15 leu2–3,112 Δtrp1 ura3–1, ade2–1 can1–100*) or BY4741 (*MATa his3Δ leu2Δ0 met15Δ ura3Δ0*) and are listed in S2 Table.

YAR015-4C was generated by genetic cross of YDZ7005–16 and SCHY58A. HPR1DGK is a diploid strain generated by mating HPR1-D2 and SCHY58A. WRIEHA and WSHEHA were generated from YBP249 by fusion in the C-terminal region of the *RIE1* or *SHE2* ORFs (respectively) of a *3HA::ADH1::kanMX6* cassette amplified by PCR from pFA6a-3HA-kanMX6 plasmid, using the primers RIE1-HA fw, RIE1-HA rv, SHE2-HA fw, SHE2-HA rv and RECOMB rv.

Media used in this study: YPAD (1% yeast extract, 2% bacto-peptone, 2% glucose, and 20 mg/ml adenine), Synthetic defined [63] (0,17% yeast nitrogen base without amino acids, 0,5% ammonium sulfate, supplemented with amino acids), synthetic complete (SC) (SD with 2% glucose, 2% galactose or 2% raffinose). Solid media was prepared by adding 2% agar.

For serial dilution growth assays, mid-log cultures were grown in selective media. Ten-fold serial dilutions of the culture were done using sterile water and 5 µl of each dilution was plated and grown for 2–3 days at 30°C. For Hpr1 depletion (HPR1DGK), 1 mM of auxin (1-Naphthaleneacetic acid, NAA, from Sigma) was added to the media. Cultures were shifted to 37ºC for 2h to analyse *dis3–1* (BDIS3–1) mutant and for 1h to analyze the *mex67–5* (WMC1-1A) under non-permissive conditions. Yeast strains were defrosted from glycerol stocks and grown at 30°C using standard procedures.

### Plasmids

Plasmids used in this study are listed in S3 Table.

pYES-NPL3, pYES-RIE1, pYES-RIM4, pYES-SBP1, pYES-SHE2 and pYES-YFP were generated by cloning of PCR cassettes amplified from their respective ORFs (or from pRS413-GAL::YRA1Δi or pWJ1344) into pYES2 multicloning site, using the listed primers: NPL3-XhoI fw, NPL3-XhoI rv, RIE1-KpnI fw, RIE1-EcoRI rv, RIM4-KpnI fw, RIM4-SacI rv, SBP1-HindIII fw, SBP1-XbaI rv, SHE2-XhoI fw, SHE2-XhoI rv, YFP fw and YFP-EcoRI rv. pYES-YRA1Δi was generated by cloning of the BamHI fragment of pRS413-GAL::YRA1Δi containing *YRA1* gene without intron into pYES2 BamHI site. pYES-RIM4Δ448–718 was generated by deletion of the last C-terminal 813 pb of *RIM4* ORF in pYES-RIM4 using a PCR product amplified from pFA6a-3HA-kanMX6 plasmid using RIM4Δ448–718-HA fw and RIM4-HA rv primers. pYES-RIE1YFP and pYES-SHE2YFP were obtained by fusion of their respective ORFs to YFP by a two-step PCR from

pWJ1344, pYES-RIE1 and pYES-SHE2 using the primers RIE1-KpnI fw, RIE1-PH rv, SHE2-KpnI fw, SHE2-PH rv, YFP fw, YFP-EcoRI rv and cloned into KpnI/EcoRI sites of pYES2. pYES-RIE1HA and pYES-SHE2HA, were generated by fusion of 3xHA in the C-terminal end of their ORFs by introducing in pYES-RIE1 and pYES-SHE2 a PCR fragment obtained from WRIEHA and WSHEHA using RIE1comp fw, SHE2comp fw and HAtag rv primers. pYES-SBP1HA, pYES-RIM4HA and pYES-RIM4Δ448–718HA were generated by fusion of 3xHA in the C-terminal end of their ORFs by introducing in their corresponding pYES vectors a PCR fragment obtained from pYES2-SHE2HA using SBP1-HA fw, SBP1-HA rv, RIM4-HA fw, RIM4-HA rv or RIM4Δ-HA rv primers. YEpDIS3 was obtained by cloning into YEp351 SacI/SphI site a PCR product containing *DIS3* ORF plus 457pb upstream and 126pb downstream, amplified from YPB249 using DIS3-SacI fw and DIS3-SphI rv primers. YEpDIS3-HA was created by cloning in YEpDIS3 3xHA tag at C-terminal end obtained by PCR from WDISHA genomic DNA using DIS3Sal fw and DIS3-HA rv primers. YEpYRA1Di was generated by cloning of the BamHI fragment of pRS413-GAL::YRA1Δi containing *YRA1* gene without intron into YEp351 BamHI site. MW90-C1A and MW90-C1B were obtained by enzymatic digestion using XbaI or BamHI (respectively) and religation of the plasmid MW90-C1. Primer sequences are listed in S4 Table.

### mRNP purification

mRNP purification from cells expressing Cbp80-proteinA was performed essentially as described [23]. Briefly, we grew cells in 3L of YPAD media at 30ºC to OD 1-1.2, collected them and washed them and resuspend them in a volume of 20mM Hepes pH7.4 1.2% PVP-40. Cells were extruded from a syringe into liquid nitrogen to quickly freeze them and grinded in a planetary mill (Pulverisette, Fritsch) in a 80 ml vase with 4 balls 3 times 500rpm 3 minutes with 1 min reverse. The jars were place in liquid nitrogen between cycles to keep the cells frozen. 2 gr of frozen cell powder were use per immunoprecipitation. Frozen cells were resuspended in nine volumes of extraction buffer (20mM Hepes pH 7.4, 110mM KOAc, 75mM NaCl, 0.1% Tween-20, 0,5% Triton X-100, 1mM PMSF, 1X protease inhibitors cocktail complete EDTA-free and 1:5000 antifoam B 1:5000 RNasein). The resulting extract was clarified by filtering it through 1.6 µm GD/X Glass Microfiber syringe filters (25mm, Whatman) and incubated for 30min at 4°C with IgG-conjugated magnetic beads. Beads were then washed four times with extraction buffer and once with 0.1 M $NH_4OAc$, 0.1mM $MgCl_2$, 0.02% Tween-20. Bound-complexes were eluted with 0.5M $NH_4OH$, 0.5mM EDTA, lyophilized and resuspended either in SDS-sample buffer for SDS-PAGE or in 25mM ammonium carbonate for mass spectrometry analysis.

### Mass spectrometry

Digestion was performed overnight at 37°C in the presence of 12.5 µg/ml of sequencing grade trypsin (Promega). Digests were analyzed by a LTQ Velos Orbitrap (Thermo Fisher Scientific) coupled to an Easy nano-LC Proxeon system (Thermo Fisher Scientific). Chromatographic separation of peptides was performed with the following parameters: column Easy Column Proxeon C18 (10 cm, 75 µm i.d., 120 Å), 300nl/min flow, gradient rising from 95% solvent A (water - 0.1% formic acid) to 25% B (100% acetonitrile, 0.1% formic acid) in 20min, then to 45% B in 40min and finally to 80% B in 10min. Peptides were analyzed in the Orbitrap in full ion scan mode at a resolution of 30,000 and a mass range of 400–1800 m/z. Fragments were obtained with a collision-induced dissociation (CID) activation with a collisional energy of 40%, an activation Q of 0.25 for 10ms, and analyzed in the LTQ. MS/MS data were acquired in a data dependent mode in which 20 most intense precursor ions were isolated, with a dynamic exclusion of 20 seconds and an exclusion mass width of 10ppm. Label Free quantitation was performed using Progenesis QI for proteomics software version 4.2 (Waters). The software was allowed to automatically align data to a common reference chromatogram to minimize missing values. Then, the default peak-picking settings were used to detect features in the raw MS files and a most suitable reference was chosen by the software for normalization of data following the normalization to all proteins method. A between-subject experiment design was chosen to create groups of four biological replicates, i.e., control (no tag), CBP80-pA wt and CPB80-pA *hpr1*Δ. MS/MS spectra were exported and searched against the SwissProt database with the *S.cerevisiae* taxonomy using

Proteome Discoverer 1.4 software (ThermoFisher Scientific) using a search engine node for an in-house Mascot server version 2.4.1 (Matrix Science). A maximum of 2 missed cleavage sites was authorized. Precursor and fragment mass tolerances were set to respectively 7 ppm and 0.5 Da. The following modifications were considered in mass calculation: oxidation (M), phosphorylations (STY), acetylation (Nterm, K), deamidation (NQ). The maximum number of missed cleavages was limited to 2 for trypsin digestion. Peptide spectrum-matches (PSMs) were filtered using a 1% False Discovery Rate (FDR). Identification results were then imported into Progenesis to convert peptide-based data to protein expression data using the Hi-3 based protein quantification method. Data were then processed using multivariate statistics to evidence differentially enriched proteins meeting the following criteria: at least two unique peptides, Fold Change higher than 2, ANOVA p-values lower than 0.05 and power higher than 0.8. Proteins significantly enriched in either CBP80-pA samples relative to the no tag control were further compared for their presence in wt vs. *hpr1Δ* mRNPs.

### Rad52-YFP foci detection

Spontaneous Rad52-YFP foci formation from mid-log growing cultures were visualized by fluorescence microscopy in a Leica DM6000 B microscope at x63 magnification, as described [64]. Cultures were fixed with 2.5% formaldehyde prior to visualization. At least 200 S/G2 cells were manually counted for each experiment.

### Nuclear protein localization

Mid-log cultures growing overnight in 2% galactose SC media were fixed and nuclei were stained with DAPI. Images of the YFP-fused proteins and DAPI were visualized and acquired using a Leica DM6 B microscope equipped with a DFC390 camera (Leica) at x100 magnification using the LAS AF software (Leica). YFP Average intensity was measured in the nucleus and the cytoplasm of each cell using Metamorph Offline version 7.8.13.0 (Molecular Devices, LLC) software. At least 150 cells where quantified per replicate.

### Chromosome spreads immunostaining of DNA-RNA hybrids

Surface spreading of yeast nuclei were performed as previously described [65]. DNA-RNA hybrids were immunostained with S9.6 antibody and chicken anti-mouse Alexa Fluor 594 conjugated secondary antibody. Nuclei were stained with DAPI. At least 100 nuclei per replicate were acquired using a Leica DM6 B microscope equipped with a DFC390 camera (Leica) at x100 magnification using the LAS AF software (Leica) to obtain the percentage of nuclei with detectable S9.6 signal.

### Fluorescence *in situ* hybridization (FISH)

mRNA export from mid-log cultures grown in 2% galactose SC media was measured as previously described [66] with minor modifications. Fixed cells were spheroplasted with 2 mg 20T zymoliase (USBiological) at 30ºC during 15 min., washed and extended in poly-L-Lysine (Sigma-Aldrich) coated slides. Samples were hybridized for 3 h with 200 ng oligodT (40xT) Cy3-labeled (Sigma-Aldrich) in hybridization buffer (2x SSC, 20% formamide, 2.5 mg/ml BSA, 10 mM VRC containing a total of 200 ng of Cy3-oligodT, 100 µg of sonicated salmon sperm DNA and 100 µg g of *E. coli* tRNA). DAPI was used to stain the nuclei. Images were acquired in a Leica DM6000 B microscope equipped with a DFC365FX camera (Leica).

### RNA isolation from yeast

RNA was extracted from mid-log cultures using acid phenol after 3 h of galactose induction, following standard procedures. Briefly, mid-log cultures were collected and washed in cold water before resuspending the cell pellet in TES buffer (10 mM Tris, 1 mM EDTA, 0.5% SDS in DEPC-treated water) and immediately mixing with acid phenol (Sigma). Samples

were incubated at 65ºC during 45 minutes with occasional vortexing. Aqueous phase was recovered after two phenol and one chloroform extraction, and RNA precipitated with sodium acetate and cold ethanol.

### Northern blot assay

RNA was separated by acrylamide gel electrophoresis and transferred to Hybond-N nitrocellulose membranes (GE Healthcare). $^{32}$P-labelled DNA probes were used. Signal was acquired using a FLA-5100 Imager Fluorescence Analyzer (Fujifilm) and quantified with MultiGauge 2.0 analysis software (Science Lab). Signal was plotted as arbitrarily units (a.u.).

### RT-qPCR

RNA was converted into DNA using the QuantiTect Reverse Transcription kit following the manufacturer's instructions. Quantitative PCR were performed at given regions using SYBR Green PCR Master Mix (Biorad) in a 7500 Fast Real Time PCR System (Applied Biosystems).

### Western blot assay

Western blots were performed on proteins extracted using 10% TCA following standard procedures. Samples were loaded onto 4–20% acrylamide gels (Bio-Rad), migrated in an SDS-containing buffer, transferred to nitrocellulose membranes and hybridized with anti-HA primary antibody or anti-PGK1 antibody, followed by anti-rabbit or anti-mouse HRP-conjugated secondary antibodies (S5 Table). Images were acquired using Amersham ImageQuant 800 (Cytiva) and quantified with Image Lab (Bio-Rad).

### Chromatin immunoprecipitation (ChIP)

Mid-log cultures were grown in 2% galactose SC prior to fixation with 1% formaldehyde. Chromatin was processed as previously described [67] with minor modifications. Cells were resuspended in lysis buffer (50 mM HEPES-KOH at pH 7.5, 150 mM NaCl, 1 mM EDTA, 1% Triton X-100, 0.1% sodium deoxycholate, 0.1% SDS, 1 mM phenylmethanesulfonyl fluoride, EDTA-free protease inhibitor) and broken with glass beads in a VXR basic (Vibrax) shaker at 4ºC. Chromatin was sonicated to an average fragment size of 400–500 bp using a Bioruptor UCD-200 (Diagenode) and immunoprecipitated with 10 µL of HA tag antibody bounded to Protein A Dynabeads (Invitrogen). Quantitative PCR were performed at given regions using SYBR Green PCR Master Mix (Bio-Rad) in a 7500 Fast Real Time PCR System (Applied Biosystems). Relative abundance of the protein for each region was normalized to their input signal.

### DNA-RNA hybrid immunoprecipitation (DRIP)

Mid-log cultures were grown overnight in 2% galactose SC media. DNA-RNA hybrids were processed as previously described [20]. Half of the samples was treated with 8µl RNase H (New England BioLabs) overnight at 37°C as negative control. Quantitative PCR were performed at given regions using SYBR Green PCR Master Mix (Bio-Rad) in a 7500 Fast Real Time PCR System (Applied Biosystems). Relative abundance of DNA-RNA hybrids in each region was normalized to their input signal.

## Supporting information

**S1 Fig. Analysis of mRNP composition in the absence of THO.** (A) mRNPs purified using immunoglobulin gamma (IgG)-conjugated beads pulling down either protein A-tagged Cbp20 or Cbp80 in a wild-type or *hpr1Δ* strain. Proteins were separated in SDS-PAGE stained with Coomassie or a Deep-Purple fluorescent dye. Plot comparing the fluorescence intensity of the mRNP components isolated from a wild-type or *hpr1Δ* strain. (B) RNA of WT strain (YBP249) overexpressing (+) or not (-) Npl3, Rim4, Sbp1 and She2 were analyzed by RT-qPCR. (C) Lysates from WT strain (YBP249)

overexpressing (+) or not (-) Rim4 or Sbp1 and from a She2-HA tagged strain (WSHEHA) overexpressing (+) or not (-) She2 were analyzed by Western blot. (D) Lysates from WT strain (YBP249) overexpressing Rim4 or She2, with (+) or without (-) RNase H overexpression were analyzed by Western blot. Mean and SD of 3 samples is plotted for (D). *p ≤ 0.05; (two-tailed Student's t-test).
(EPS)

**S2 Fig. Analysis of Rie1 and Dis3 overexpression** (A) Diagram of genomic sequence contained in MW90-Clone 1 plasmid (C1), the deletions analysed (C1 del.A and C1 del.B) and YEpDIS3 plasmid (*DIS3* o.e.). (B) Diagram of genomic sequence contained in MW90-Clone 2 plasmid (C2) and pYES2-RIE1 plasmid (*RIE1* o.e.). (C) Lysates from WT strain (YBP249) overexpressing (+) or not (-) Rie1 or Dis3 were analyzed by Western blot. (D) Lysates from WT strain (YBP249) overexpressing Rie1 or Dis3, with (+) or without (-) RNase H overexpression were analyzed by Western blot.
(EPS)

**S1 Table. Proteins increased/decreased in mRNPs isolated in the absence of THO.**
(XLSX)

**S2 Table. Yeast strains used in this study.**
(PDF)

**S3 Table. Plasmids used in this study.**
(PDF)

**S4 Table. Primers used in this study.**
(PDF)

**S5 Table. Antibodies used in this study.**
(PDF)

**S1 Information. Raw data.**
(XLSX)

## Acknowledgments

We are grateful to M. Oeffinger for the Cbp80 and Cbp20 tagged strains.

## Author contributions

**Conceptualization:** Ana G. Rondón, Andrés Aguilera.

**Data curation:** Guillaume Chevreux.

**Formal analysis:** José Antonio Mérida-Cerro, Guillaume Chevreux.

**Funding acquisition:** Benoit Palancade, Ana G. Rondón, Andrés Aguilera.

**Investigation:** José Antonio Mérida-Cerro, Guillaume Chevreux, Ana G. Rondón.

**Supervision:** Benoit Palancade, Ana G. Rondón, Andrés Aguilera.

**Validation:** Benoit Palancade, Ana G. Rondón.

**Visualization:** José Antonio Mérida-Cerro.

**Writing – original draft:** Ana G. Rondón.

**Writing – review & editing:** José Antonio Mérida-Cerro, Benoit Palancade, Ana G. Rondón, Andrés Aguilera.

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
