## [Decision Letter · Decision Letter 0]

PGENETICS-D-24-01326

Different alterations in mRNP composition trigger an increase in harmful R-loops

PLOS Genetics

Dear Dr. Rondon,

Thank you for submitting your manuscript to PLOS Genetics. After careful consideration, we feel that it has merit but does not fully meet PLOS Genetics's publication criteria as it currently stands. Therefore, we invite you to submit a revised version of the manuscript that addresses the points raised during the review process.

Please submit your revised manuscript within 60 days Feb 16 2025 11:59PM. If you will need more time than this to complete your revisions, please reply to this message or contact the journal office at plosgenetics@plos.org. Please include the following items when submitting your revised manuscript:

We look forward to receiving your revised manuscript.

Kind regards,

Dmitry A. Gordenin, Ph.D.

Academic Editor

PLOS Genetics

Monica Colaiácovo

Section Editor

PLOS Genetics

Aimée Dudley

Editor-in-Chief

PLOS Genetics

Anne Goriely

Editor-in-Chief

PLOS Genetics

**Journal Requirements:**

At this stage, the following Authors/Authors require contributions: Benoit Palancade. Please ensure that the full contributions of each author are acknowledged in the "Add/Edit/Remove Authors" section of our submission form.

The list of CRediT author contributions may be found here: https://journals.plos.org/plosgenetics/s/authorship#loc-author-contributions

**Reviewers' comments:**

Reviewer's Responses to Questions

Reviewer #1: Title: Different alterations in mRNP composition trigger an increase in harmful R-loops.

Manuscript# PGENETICS-D-24-01326

Authors: Mérida-Cerro et al.

In this manuscript, authors have investigated the role of mRNP in R-loop homeostasis and genome instability. They have identified three RNA-binding proteins (Rie1, Rim4 and She2) and an exosome component Dis 4 and have shown that the stoichiometric levels of these proteins affects the accumulation of R-loops and DNA damage. Authors uncovered that mRNP assembly is a tightly regulated process whose perturbation leads to R-loop-mediated DNA damage and genomic instability. The objectives of this investigation are logical, and experimental techniqes are thorough and technically competent. The experimental data in the manuscript are robust, statistically analyzed accurately, and support their conclusions. The findings of this manuscript should be of broad interest to the scientific community especially those working in the field of genomic stability.

Listed below are comments that would be useful if the authors have the data or can easily address them experimentally:.

1. Previous studies from these authors have shown that Yra1 overexpression stabilizes R-loops (Garcia-Rbio et al. 2018). In this study authors have combined Yra1 overexpression with Hpr1 deletion (Fig. 1), which may theoretically leads to increased accumulation levels of R-loops. Have authors examined the levels of R-loops in hpr1-null strains with or without Yra1 overexpression? If possible it will be useful to define the threshold of R-loops that is tolerated by the cell and/or the levels beyond that may cause DNA damage and genomic instability.

2. Authors in Figure 1D have examined the Rad52 foci upon overexpression of Rim4 and She2 in the presence or absence of RNH. This experiments was performed to establish the role of R-loops in DNA damage for experimental data shown in Figure 1C. Have authors examined the suppression of Rad52 foci by RNH in strains overexpressing other candidate genes such as Npl3, Yra1and Sbp1? It is unclear whether this phenotype is specific to these two genes (Rim4 and She2) or it is a general phenomenon.

3. Overexpression of DIS3 (Figure 3) and its loss of fundction mutant dis3-1 (Figure 4) show similar phenotypes as in both cases R-loops accumulation was observed. Discussion about this will be useful to the readers.

4. Authors have made a novel observation for Rim4 where they have discovered a novel prion domain that may have a role in R-loops homeostasis. Authors concluded that induction in the levels of Rim4 leads to aggregate formation that makes Rim3 non-functional. It is not clear whether this prion domain in Rim4 has direct role in R-loops accumulation. Perhaps authors could examine any other protein that has prion domain but no role in mRNP assembly and R-loops homoestatis to clearly establish the role of prion domain or aggregate formation in R-loops mediated genome instability.

5. Accumulation of R-loops in dis3-1 and rim4del strains correlated with increased Rad52 foci, a measure for DNA damage that is suppressed by RNH (Figure 4). Have authors examined the levels of ssDNA in these strains as an independent measure to further establish the role of R-loops?

6. In Figure 5, authors show that Rie1 when overexpressed recruited to nucleus but its nuclear localization was not detected when expressed at its endogenous levels. This could be due to limits of assays used or limited number of genomic regions examined. Although not needed for this paper, it would be interesting to examine genomic localization of Rie1 by other assays as ChIP-seq.

7. Authors have examined R-loops at certain genomic loci upon Dis3 overexpression. However, it is unclear whether Dis3 itself associate with these regions. Will be interesting to define the genomic regions bound by Dis3 as this may help better understand the functional role of Dis3.

8. In the abstract specify that Dis3 is an exosome component

9. Previous studies from human cells and budding yeast have implicated a role for R loops at centromere and chromosome segregation. Authors should mention and cite these references (Guinta et al and Mishra et al).

10. Will be useful to describe a bit more about the relevance of these studies to human disease for a broad audience.

Reviewer #2: Co-transcriptional assembly of RBPs and nascent RNAs are crucial for proper mRNA packaging, which if not completed in a timely and efficient manner, may lead to R-loop formation. The THO complex plays a supporting role in packaging (i.e., the THO complex is non-essential in budding yeast) and when deleted leads to increased detection of R-loops. Merida-Cerro et al., using a combination of proteomics and genetics, identified four additional factors, including one exosome component and three RBPs, whose alteration similarly increased R-loop formation a measured by S9.6 based methods. This manuscript proposes that these changes are “triggered” by alterations in mRNP composition, but the paper lacks any direct evidence of this and is solely dependent on the observation that overexpression or mutation of these factors increases R-loop formation. Merida-Cerro et al. do present experiments that suggest the involvement of these factors, but they lack mechanistic insight, particularly regarding the fact that changes in mRNP composition proceed and trigger R-loop formation. Furthermore, there is insufficient data to refute the indirect impact of altering these factors has on cells, which are expected to influence gene expression regulation and potentially alter the expression levels of many genes, including R-loop-related factors. From the data presented, it is clear the authors have identified multiple factors that influence R-loop formation and DNA damage (as measured by Rad52-foci), but the mechanisms and cause of these phenotypes is not clear, nor match the title and narrative presented. Overall, the manuscript requires significant additional experimentation to meet the claims in the title and text or significant revision of the text to match the data presented.

Major points:

1. Proteomics data serves as the basis for proposing mRNP composition changes are linked to R-loop formation. Broadly, if these interactions are occurring in the context of R-loops and the site of transcription, it would be more appropriate to discern mRNPs from nascently forming mRNPs in the context of the DNA template. As it stands, the use of the general term mRNP is confusing, since mRNPs exist throughout the cell at all stages of gene expression and are not a single entity.

In addition, there are multiple aspects of the MS data that need to be addressed by the authors. First, it is unclear what is being isolated from each pullout and being compared. For example, the authors identify both RNA polymerase I and II subunits in their hpr1 MS data, suggesting that stalled transcription complexes in association with DNA (e.g., heavy chromatin) may be isolated (PMID:18957205). As such, specific interactions occurring in the context of R-loops and the site of transcription induced in the hpr1 mutant could be observed (e.g., Yra1 binding DNA-RNA hybrids), which are not reflective of mRNP composition. In contrast, in WT cells it may be mature mRNPs that are elsewhere in the gene expression pathway, including cytoplasmic mRNAs that have yet to be translated, which are being isolated. These being messages that would be expected to have Yra1, Npl3, and other nuclear RBPs removed, so they would appear to be reduced in comparison to those isolated from hpr1 cells.

Second, an increase in the levels of nuclear mRNPs alone, independent of R-loops, at steady-state due to altered kinetics of mRNA processing and export in a hpr1 strain would also give rise to changed interaction profiles. Increased levels of nuclear mRNPs are in fact noted by the authors as the reason for increased levels of NPC components seen in the MS data. Yra1 and Npl3 were recently reported as common components of nuclear mRNA complexes (PMID: 37963019 and 37399331), so it is expected that these factors would be enriched.

Third, the impact of the hpr1 mutation may induce changes in protein-protein and ncRNA directed interactions with Cbp80 that is not reflective of mRNPs. It can also be that the cellular transcriptome is different in an hpr1 mutant with quality control features engaged due to processing defects in the mutant. All these possible changes would alter steady-state protein-protein and protein-RNA complex interactions that are being detected by MS and interpreted by the authors as reflecting mRNPs.

To address these possibilities, and other indirect effects, it is recommended to repeat the MS (or perform IPs and western blotting) under conditions of RNase H OE, RNAse T1, and RNase A treatment to further address the nature of these interactions. In addition to incorporating a more granular discussion of what the data may reflect (see comments above).

2. The proteomics data shows THO complex components are loss by hpr1∆ in a Cbp80 pullout, yet the established TREX component Yra1 is increased. This is the opposite of published data (PMID: 37963019), where Cbp80 pulldown in a tho2 deletion strain showed a reduced amount of Yra1 in isolated mRNPs. Related impacts on the recruitment of Yra1 to gene loci has also been reported for THO complex mutants, including hpr1 (PMID: 12417727). One possible explanation, related to point #1, is that what is detected in association with Cbp80 includes materials other than “mRNPs”. This may include proteins (e.g. Yra1) bound to DNA:RNA hybrids, which are increased in the hpr1∆ strain. The authors should address this literature in the interpretation of their data.

3. With respect to the reason for R-loop formation, the manuscript does not prove a mechanism for any of the proteins of focus. In some cases, the data and ideas put forth even seem contradictory. For example, the authors present evidence that overexpression of Rim4 leads to increased R-loops, but this is attributed to the loss of functional Rim4 due to protein aggregation (Fig. 4F-H). Yet, Rim4 binding to mRNPs is enhanced in the hpr1∆ strain, which exhibits more R-loops. Similar, the authors present data for She2 suggesting that chromatin binding is dependent on transcription and R-loops. This implies that She2 binding increases in the hpr1∆ strain due to R-loop accumulation, contradicting the notion that alterations in She2 stoichiometry within the mRNP trigger R-loop formation.

4. There are reported issues with using S9.6 in localization studies (PMID: 33830170) and the detection of hybrids vs. dsRNA should be addressed in the text and with RNase T1 treatment. In addition, the S9.6 staining is often a single focus on the edge of the DAPi stain, which is very reminiscent of the position of the yeast nucleolus. The authors should check for co-localization with a nucleolar marker, since if this is the nucleolus, the discussion of this data in relation to mRNPs would change, as this may reflect R-loops forming at highly transcribed rDNA loci.

5. Related to #1 and #2, experimental conditions, including lysis buffer composition and cryo-grinding procedures, may impact the isolation of chromatin associated fractions containing DNA:RNA hybrids and bound RBPs vs soluble mRNPs. The paper lacks a detailed description of the pulldown procedure in the materials and methods that would inform this point, this detail must be added. Another noted detail missing within the methods is the growth temperature of the mex67-5 strain for FISH. It is recommended that the authors ensure sufficient details are provided in the methods section related to all experiments performed.

Other points:

1. No representative images of mutants Rad52 foci data is provided, which should be added.

2. No data is shown to address the level of overexpression of the various proteins. This makes the strength, and importantly the lack of a phenotype, difficult to address. The authors need to confirm that they are overexpressing these proteins.

3. Images lack scale bars.

4. Image quality and brightness, in particular Figure 5, are an issue as presented. Fig 5 A appears saturated and is difficult to interpret, also the defect of a mex67-5 mutant appears to be a lack of a dT signal and a very weak single focus, which is not consistent with literature.

5. Figure 2A is difficult to assess, serial dilutions should be used in place of this figure panel.

6. Yeast strain list is written as supplemental table 1, but this is supplemental table 2. Plasmid list is written as supplemental table 2 but this is table 3.

Reviewer #3: In the manuscript by Mérida-Cerro et al. entitled “Different alterations in mRNP composition trigger

an increase in harmful R-loops” the authors employ a combination of biochemical and genetic approaches to identify proteins that could influence R-loop accumulation. The studies identify four proteins for which the authors probe mechanisms influencing R-loops.

The title could use some modification as “Different alterations” is a rather confusing term. The authors suggest distinct mechanisms by which these RNA binding and RNA regulatory factors could impact R-loops, so perhaps the title could more accurately reflect this conclusion.

As another point of editing, the authors need to ensure consistent use of the term “RNA exosome” rather than the shortened term “exosome”. The explosion of studies of extracellular vesicles termed ‘exosomes’ has led to MUCH confusion in the literature, so the term used here must be RNA exosome. This change should be made throughout the manuscript even when the word RNA seems a bit redundant.

The complementary biochemical and genetic data is a strength of the study. There are some additional controls and some missed opportunities to bolster the authors’ conclusions about mechanism.

Detailed Comments:

Figure 1 presents results of the mass spec analysis to identify candidates and then employs genetics as a next step. The experiments with the Rad52 foci (Figure 1C and Figure 1D) could benefit significantly from accompanying immunoblots to assess the amount of overexpression achieved. This point would be informative for the data shown in Figures 1B and 1C albeit the authors would likely need to employ epitope tags due to the limited antibody availability in yeast and the fold overexpression relative to endogenous would most likely not be feasible to determine; however, this point is critical for Figure 1D where the authors add in overexpression of RNAseH. The authors need to demonstrate that the level of expression of each of the proteins is the same in the cells plus and minus RNH expression to confirm that the rescue of Rad52 foci is indeed due to RNAseH activity. The authors could also overexpress catalytically inactive RNAseH, but this experiment, while useful, would not eliminate the possibility that overexpression of another protein simply decreases the level of the Rim4 and/or She2 as compared to conditions where RNAseH is not overexpressed.

The Rad52 foci assay is nice to use for a measure of DNA damage; however, the authors never present any of the raw data (micrographs) for any of these experiments, which is a bit odd. Several of the figures present quantitation of such experiments, so including the images for at least one set of data would be nice.

Minor point- Figure 1A could be labeled so that a reader knows that the comparison is between wildtype/control and hpr1Δ cells. Currently, the reader has to consult the text or Figure Legend to understand what is being compared in this figure. A more informative Y-axis lab could easily rectify this minor point.

The authors then performed a genetic screen to identify genes that when overexpressed in hpr1Δ cells as compared to control cause a growth defect. The cell streaks shown in Figure 2A are not terribly convincing and a more conventional figure would be the serial dilution and spotting assay that the authors employ for all other similar experiments. The authors also provide a bit too much granular detail about their work to narrow the suppressors. A typical way to display these data would be to include the original suppressor plasmids in Figure 2D or if complicated by the slight differences between the way the screen was performed (Hpr1 degron/multicopy library) and how the experiment in Figure 2D is performed (hpr1Δ cells and galactose induction), then simply provide the growth assay with the multicopy suppressors- seems like this could be done in the hpr1Δ cells as a serial dilution assay. The information about the actual suppressor plasmids seems like that could be omitted or if the authors wish presented in Supporting Information- this applies to Figures 2C and 2E and the associated text in the manuscript.

Figure 2F suffers from the same issue as Figure 1D in that demonstrating similar levels of overexpression achieved in the absence and presence of Dis3 and Rie1 overexpression is critical to draw the conclusion linking these factors to R-loop biology.

Minor point- In the current flow, Figure 2E seems out of order as Figure 2D presents Rie1 overexpression.

Figure 3 examines R-loops using the S9.6 antibody. The representative figures shown are fine for a single cell, but a more convincing experiment would include a field of cells to give some impression of distribution of R-loops as quantified in the data presented in Figure 3B. This may or may not be possible with the chromosome spreads. In addition, a positive control such as Yra1 or simply a Tho mutant is missing so the reader has no context for whether the change from 17-21% to 9% is of a similar magnitude to other established regulators of R-loop biology. The figure also gives no indication and the information also is not in the Figure Legend- so one has to consult the Methods for this information. The authors should ideally indicate the number of cells analyzed in the figure, but perhaps the Figure Legend would be sufficient. Having to look to the Methods is not ideal.

Figure 4 provides the nice complement of examining whether loss of function mutations cause the same effects as the overexpression of each protein analyzed. The authors then go on to explore a variant of Rim4 and assess whether the IDR domain of Rim4 is required to confer the R-loop accumulation observed upon overexpression of Rim4 (Figures 4F-H). For these experiments, a domain structure showing precisely what domain is removed would have been helpful. Perhaps this should be a separate figure. For these data, demonstrating that the steady-state protein level is similar for the Rim4 and the rim4Δ271 is absolutely critical to support the authors conclusions. Thus, immunoblotting to assess and compare these protein levels is absolutely essential to support the conclusions drawn based on the data shown in Figures 4F-H.

At some point, the authors miss a bit of an opportunity to ascribe more mechanism to the Dis3 overexpression because they could have opted to overexpress a variant lacking the catalytic exo and/or endo activity to understand if the effects observed are due to the catalytic activity or physical interactions with other RNA exosome subunits.

Really minor points:

The word “since” generally implies time- I have been working on this manuscript since March”- whereas the word “as” is preferable when there is no element of time- such as on p. 6 “Since Yra1 is

overrepresented in hpr1Δ mRNPs (Fig 1A),..” should really ore correctly be “As Yra1 is overrepresented in hpr1Δ mRNPs (Fig 1A),..”

There are a few other grammatical errors of odd word choices that make following the text as bit challenging:

p. 6 “Yra1 expression is tightly regulated in the cell, being its overexpression toxic (25, 26),..”- being is an odd word choice

p. 7 “…being RIM4 and SBP1 already toxic in wildtype cells (Fig 1B).”- being is an odd word choice

p. 7 The term “cell sickness” is more of lab speak- we would tend to use “poor or slow growth”

p. 7 “We performed a screening for genes that cause cell sickness..” should be “we performed a

SCREEN for genes that cause decreased cell growth..” or something similar- correcting both the word screening to screen and the cell sickness.

p. 12 “at GCN4 and 18S rDNA, overexpression of rim4Δ271 presented wild-type levels in both genes (Fig 4H).” the term “wild-type levels” does not really make sense here.

**Have all data underlying the figures and results presented in the manuscript been provided?**

Reviewer #1: Yes

Reviewer #2: Yes

Reviewer #3: Yes

PLOS authors have the option to publish the peer review history of their article (what does this mean? ). If published, this will include your full peer review and any attached files.

**Do you want your identity to be public for this peer review?** For information about this choice, including consent withdrawal, please see our Privacy Policy .

Reviewer #1: No

Reviewer #2: No

Reviewer #3: No

**Figure resubmission:**
---

## [Decision Letter · Decision Letter 1]

Dear Dr Rondon,

We are pleased to inform you that your manuscript entitled "Cellular imbalance of specific RNA binding proteins associates with harmful R-loops" has been editorially accepted for publication in PLOS Genetics. Congratulations!

Before your submission can be formally accepted and sent to production you will need to complete our formatting changes, which you will receive in a follow up email. Please be aware that it may take several days for you to receive this email; during this time no action is required by you. Please note: the accept date on your published article will reflect the date of this provisional acceptance, but your manuscript will not be scheduled for publication until the required changes have been made. We also ask you to please address the minor comment made by Reviewer #2.

Yours sincerely,

Dmitry A. Gordenin

Associate Editor

PLOS Genetics

Monica P. Colaiácovo

Section Editor

PLOS Genetics

Aimée Dudley

Editor-in-Chief

PLOS Genetics

Anne Goriely

Editor-in-Chief

PLOS Genetics

Comments from the reviewers (if applicable):

Reviewer's Responses to Questions

**Comments to the Authors:**

Reviewer #1: The authors have addressed my comments

Reviewer #2: The text edits, updates to figures, and addition of new data is appreciated and addresses the majority of the concerns raised. A few small edits are requested in line with the comments made during the first round of review:

1. Change the abstract to remove mention of stoichiometry. It is more accurate to use the term "expression levels" and suggest the following:

"Here, we identify three RNA-binding proteins, Rie1, Rim4 and She2, whose expression levels are important to limit R-loop accumulation and, thus, to prevent DNA damage. Interestingly, Rim4 and She2 are overrepresented in Cbp80 containing complexes formed in the absence of THO."

2. Similar changes are requested in the intro and results text, as follows:

To understand how mRNP assembly restricts R-loops, we analysed Cbp80-containing complexes formed in the absence of the THO complex and found an increased presence of various RBPs and nuclear pore components. Moreover, we noticed that over expressing the RBPs Rie1, Rim4 and She2 or the activity of the Dis3 RNA exosome subunit promotes R-loop accumulation in the cell, consequently causing DNA damage. Therefore, the proper cellular expression levels of RBPs is crucial for preventing R-loop-mediated genome instability.

Taken together, our results imply that altering the cellular levels of Rim4, Rie1, She2 or the RNA exosome subunit Dis3, causes R-loop accumulation. Interestingly, whereas Rim4 and Rie1 seem to have a general effect, She2 and Dis3 action is preferentially directed to specific genes whose transcripts are targets of these factors.

Reviewer #3: The authors have addressed the points raised in the previous review, including adding new experimental data to support their conclusions.

**Have all data underlying the figures and results presented in the manuscript been provided?**

Reviewer #1: Yes

Reviewer #2: Yes

Reviewer #3: Yes

PLOS authors have the option to publish the peer review history of their article (what does this mean? ). If published, this will include your full peer review and any attached files.

**Do you want your identity to be public for this peer review?** For information about this choice, including consent withdrawal, please see our Privacy Policy .

Reviewer #1: No

Reviewer #2: No

Reviewer #3: No

**Data Deposition**

http://datadryad.org/submit?journalID=pgenetics&manu=PGENETICS-D-24-01326R1

**Press Queries**

---

## [Editor Report · Acceptance letter]

PGENETICS-D-24-01326R1

Cellular imbalance of specific RNA binding proteins associates with harmful R-loops

Dear Dr Rondón,

We are pleased to inform you that your manuscript entitled "Cellular imbalance of specific RNA binding proteins associates with harmful R-loops" has been formally accepted for publication in PLOS Genetics! Your manuscript is now with our production department and you will be notified of the publication date in due course.

With kind regards,

Anita Estes

PLOS Genetics

On behalf of:
